# Identification of Differentially Expressed Genes and microRNAs in the Gray and White Feather Follicles of Shitou Geese

**DOI:** 10.3390/ani14101508

**Published:** 2024-05-20

**Authors:** Pengyun Guo, Junpeng Chen, Lei Luo, Xumeng Zhang, Xiujin Li, Yunmao Huang, Zhongping Wu, Yunbo Tian

**Affiliations:** 1College of Animal Science & Technology, Zhongkai University of Agriculture and Engineering, Guangzhou 510225, China; guopengyun@zhku.edu.cn (P.G.); luolei2112322020@163.com (L.L.); zhangxumeng@zhku.edu.cn (X.Z.); lixiujin996@126.com (X.L.); huangyunmao@163.com (Y.H.); tyunbo@126.com (Y.T.); 2Shantou Baisha Research Institute of Original Species of Poultry and Stock, Shantou 515800, China; chenjunpeng02@163.com

**Keywords:** *Anser cygnoides*, Chinese domestic goose, plumage color, transcriptome sequencing

## Abstract

**Simple Summary:**

Shitou geese are the exclusive representative of a large-sized goose breed in China, typically exhibiting gray feathers but also possessing a recessive white plumage. However, the precise mechanism underlying the formation of this white plumage coloration in Shitou geese remains unclear. Here, a transcriptome sequencing analysis was conducted to identify candidate genes and microRNAs (miRNAs) that may be associated with the white plumage color in Shitou geese. A number of pigmentation genes and miRNAs were identified as candidates that may regulate white plumage color formation in Shitou geese. This result may be important for understanding the genetic mechanism underlying the formation of white plumage color in Shitou geese.

**Abstract:**

The Shitou goose, a highly recognized indigenous breed with gray plumage originating from Chaozhou Raoping in Guangdong Province, China, is renowned for being the largest goose species in the country. Notably, during the pure breeding process of Shitou geese, approximately 2% of the offspring in each generation unexpectedly exhibited white plumage. To better understand the mechanisms underlying white plumage color formation in Shitou geese, we conducted a comparative transcriptome analysis between white and gray feather follicles, aiming to identify key genes and microRNAs that potentially regulate white plumage coloration in this unique goose breed. Our results revealed a number of pigmentation genes, encompassing *TYR*, *TYRP1*, *EDNRB2*, *MLANA*, *SOX10*, *SLC45A2*, *GPR143*, *TRPM1*, *OCA2*, *ASIP*, *KIT*, and *SLC24A5*, which were significantly down-regulated in the white feather follicles of Shitou geese. Among these genes, *EDNRB2* and *KIT* emerged as the most promising candidate genes for white plumage coloration in Shitou geese. Additionally, our analysis also uncovered 46 differentially expressed miRNAs. Of these, *miR-144-y* may play crucial roles in the regulation of feather pigmentation. Furthermore, the expression of *novel-m0086-5p*, *miR-489-y*, *miR-223-x*, *miR-7565-z*, and *miR-3535-z* exhibits a significant negative correlation with the expression of pigmentation genes including *TYRP1*, *EDNRB2*, *MLANA*, *SOX10*, *TRPM1*, and *KIT*, suggesting these miRNAs may indirectly regulate the expression of these genes, thereby influencing feather color. Our findings provide valuable insights into the genetic mechanisms underlying white plumage coloration in Shitou geese and contribute to the broader understanding of avian genetics and coloration research.

## 1. Introduction

The diversity of feather colors makes birds the most colorful terrestrial vertebrates and plays an important role in attracting individuals of the opposite sex and avoiding predators [1,2]. In the poultry industry, plumage color is one of the most important varietal characteristics and economic traits, which has long been the focus of poultry science research. White plumage is highly preferred by meat-type poultry producers because of its ease of cleaning compared with pigmented feathers [3,4]. Therefore, white feathers are a common phenomenon in commercial meat-type poultry breeds, such as AA (Arbor) and Pekin ducks, although local poultry comes in a variety of feather color types.

The down feathers of waterfowl, particularly geese, possess high economic value in the light industry because of their warmth, hydrophobicity, and lack of odor [5]. The goose, as a typical type of waterfowl, occupies a crucial position in the poultry industry. The white down feathers of geese are usually more valuable than gray feathers in the light industry because they are suitable for any color fabric. Moreover, the flight feathers of geese, especially the white feathers, are the best material for sports goods such as badminton. Geese were domesticated from two wild species. Asian breeds (excluding the Yili goose) were domesticated from the swan goose (*Anser cygnoides*), known as the Chinese domestic goose, and European breeds from the greylag goose (*Anser anser*), known as the European domestic goose. The wild geese that are believed to be the ancestors of domestic geese generally present gray plumage, while those of domestic geese have both white and gray feathers. China has abundant resources of local goose breeds, and most of the local goose varieties have white feathers, such as the Zi goose, the Wanxi white goose, the Zhedong white goose, and the Sichuan white goose.

Shitou geese (*A. cygnoides*), originating from Chaozhou Raoping in Guangdong Province, are the only large-size and major goose species in China [6]. Shitou geese usually have gray feathers; however, in the pure breeding process of gray Shitou geese, approximately 2% of white plumage mutation geese will appear in each generation, which can be inherited stably (Figure 1). In general, the feather colors of geese are mainly determined by the distribution of melanin type and density, which is produced by melanocytes within feather follicles [2]. The deposition of melanin is a multifaceted process that can be broadly divided into three distinct stages as follows: melanocyte development, pigment production, and pigment distribution [7]. To date, there have been a number of reports on the functional genes of goose feather color. Wang et al. suggested the mutations of *TYR* and *MITF* were significantly associated with plumage color phenotypes in domestic geese [8]. Xi et al. discovered a significant correlation between the presence of a 14 bp insertion mutation in exon 3 of the *EDNRB2* gene and the white feather phenotype observed in Sichuan Gang geese [5]. Yang et al. and Ouyang et al. also reported an association between the same 14 bp insertion mutation in the *EDNRB2* gene and the white feather phenotype found in swan geese [9,10]. Ren et al. screened 26 genes related to goose feather color formation, including *KITLG*, *MITF*, *TYRO3*, and *KIT* [11]. Wen et al. identified a deletion mutation spanning 18 bp within the intron region of the *KIT* gene that is likely responsible for white feather color in geese [12]. However, the key genes responsible for the formation of white feathers in Shitou geese have not been reported.

MicroRNAs (miRNAs) are small non-coding RNA molecules, approximately 22 nucleotides (nt) in length, that have crucial roles in many biological processes [13]. More recently, accumulating evidence has revealed that miRNAs also play an essential role in pigmentation by regulating melanogenesis gene expression [14]. For example, Liu et al. demonstrated that *miR-380-3p* regulates melanogenesis by targeting *SOX6* in melanocytes from alpacas (*Vicugna pacos*) [15]. Wu et al. reported that *MiR-27a* regulates *WNT3A* and *KITLG* expression in Cashmere goats with different coat colors [16]. Dong et al. verified that *miR-206* plays a regulatory role in skin color pigmentation by targeting the *Mc1r* gene in Koi Carp (*Cyprinus carpio* L.) [17]. Li et al. indicated that *microRNA-200a* regulates skin pigmentation by targeting *WNT5A* and *FZD4* in Cashmere goats [18]. So far, the existence of miRNA regulating the formation of white feathers in Shitou geese has not been reported yet.

To better understand the mechanisms underlying white plumage color formation in Shitou geese, we screened differentially expressed genes and miRNAs in feather follicles from gray-feathered and white-feathered Shitou geese by RNA sequencing. Several pigmentation genes and miRNAs were identified as candidates that may regulate white plumage color formation in Shitou geese. These findings provide basic information for elucidating the genetic mechanism underlying the formation of white plumage color in Shitou geese.

## 2. Materials and Methods

### 2.1. Sample Collection

In this study, feather follicle tissues were collected from three gray-feathered and three white-feathered Shitou geese (female, 60 days). All samples were obtained from the national breed conservation farm of Shitou geese at the Shantou Baisha Research Institute of Original Species of Poultry and Stock in Guangdong Province (Shantou, China). The neogenesis of wing feathers was carefully drawn off using tweezers, and subsequently, the hair follicles were sectioned into cryopreserved tubes for immediate placement in liquid nitrogen to ensure rapid freezing preservation.

### 2.2. Total RNA Extraction, cDNA Library Preparation, and Transcriptome Sequencing

Feather follicle tissues were ground in liquid nitrogen, and total RNA was extracted using a Trizol reagent kit (Invitrogen, Waltham, MA, USA) following the manufacturer’s protocol. RNA quality was monitored by agarose gel electrophoresis and an Agilent 2100 Bioanalyzer (Agilent Technologies, Santa Clara, CA, USA). After total RNA was extracted, mRNA was enriched by Oligo (dT) beads. Subsequently, the enriched mRNA was fragmented into short fragments using fragmentation buffer and reverse transcribed into cDNA using a NEBNext Ultra RNA Library Prep Kit for Illumina (NEB #7530, New England Biolabs, Ipswich, MA, USA). The purified double-stranded cDNA fragments underwent end repair, the addition of A base, and ligation to Illumina sequencing adapters. The ligation reaction was then purified with AMPure XP Beads (1.0×). Ligated fragments were subjected to size selection via agarose gel electrophoresis and polymerase chain reaction (PCR)-amplified. Finally, the resulting cDNA library was sequenced to generate paired-end reads (150 bp) using Illumina Novaseq6000 by Gene Denovo Biotechnology Co., Ltd. (Guangzhou, China) following the manufacturer’s instructions.

### 2.3. Small RNA Library Construction and Deep Sequencing

After total RNA was extracted by the Trizol reagent kit (Invitrogen, Waltham, MA, USA), an NEB Next Multiplex Small RNA Library Prep Kit (Illumina Inc., San Diego, CA, USA) was used for small RNA library construction according to the manufacturer’s instructions. The RNA molecules in a size range of 18–30 nt were enriched by polyacrylamide gel electrophoresis (PAGE). Then, 3′ adapters were added, and the 36–44 nt RNAs were enriched. Next, 5′ adapters were ligated to the RNAs as well. The ligation products were reverse transcribed by PCR amplification, and the 140–160 bp size PCR products were enriched to generate a cDNA library and sequenced using Illumina Novaseq6000 by Gene Denovo Biotechnology Co. (Guangzhou, China).

### 2.4. Bioinformatic Analyses of Transcriptomes

To obtain clean reads, the adaptor sequences, reads with more than 10% of unknown nucleotides, and reads containing more than 50% of low-quality (Q-value ≤ 20) bases were eliminated from the raw data using fastp software v0.18.0 [19]. Subsequently, the clean reads were mapped to the *Anser cygnoides* reference genome (GCF_000971095.1) using HISAT2 v2.2.1 software [20]. The counts of all annotated genes were calculated using STRingtie v2.2.0 [21] and featureCounts [22]. The expression levels of each gene were calculated using fragments per kilobase of transcript per million mapped fragments (FPKM) [23]. Differentially expressed genes (DEGs) between gray and white feathers were then determined using DEseq2 v1.44.0 software, under a false discovery rate (FDR) below 0.05 and |log2FC| > 1 [24].

### 2.5. Processing and Analysis of Small RNA Sequences

To obtain clean tags, reads with low quality (Q-value ≤ 20) or containing unknown bases, reads with adapters, and reads shorter than 18 nt were further filtered from the raw data. All the clean tags were aligned with small RNAs in the GeneBank database (Release 209.0) and the Rfam database (Release 14.1) to identify and remove rRNA, scRNA, snoRNA, snRNA, and tRNA [25,26]. All the clean tags were also aligned with a reference genome. Tags mapped to exons or introns and repeat sequences were also removed. All the remaining clean tags were then searched against the miRBase database (Release 22) to identify known miRNAs [27]. The novel miRNA candidates were identified by mirdeep2 v0.1.3 software based on the default parameters [28]. The miRNA expression level was calculated and normalized to transcripts per million (TPM). Differentially expressed miRNAs (DEMs) were analyzed based on |log2FC| > 1 and *p*-value < 0.05 using edgeR software v4.2.0 [29]. Finally, target genes for DEM sequences were predicted using miRanda (version 3.3a) and TargetScan (version 8.0) software based on default parameters, and the intersection of the results was chosen as predicted miRNA target genes [30,31].

### 2.6. Construction of a miRNA-mRNA Interaction Network

We employed SPSS 26.0 software (IBM, Armonk, NY, USA) to conduct a two-way ANOVA analysis on the expression data of DEGs and DEMs across six samples in order to determine their correlation. Following the analysis, we identified data with extremely significant negative correlations (*p* < 0.01). Subsequently, utilizing the correlation coefficients between them, we constructed a miRNA-mRNA network diagram using Cytoscape v3.10.2 software to elucidate the relationships between DEGs and DEMs [32].

### 2.7. Quantitative Real-Time PCR Validation

To validate the differential expression of genes and miRNAs identified through RNA-Seq, the expression levels of four crucial pigmentation genes in the melanogenesis pathway, specifically, *TYR*, *EDNRB2*, *SOX10*, and *KIT*, were quantified using quantitative polymerase chain reaction (qPCR). Additionally, the expression of four miRNAs with the highest expression levels in the feather follicles, namely, *miR-451-x*, *miR-144-y*, *miR-204-x*, and *miR-196-x*, were also assessed via qPCR. mRNA primers were designed by the Primer-BLAST tool on the NCBI website (https://www.ncbi.nlm.nih.gov, accessed on 5 January 2024), and miRNA primers were designed using miRNA Design v1.01 software (Vazyme Biotech Co., Ltd., Nanjing, China). All primers were synthesized by Bioengineering Co., Ltd. (Shanghai, China). For mRNA, RNA samples were reverse transcribed into cDNA using the StarScript III All-in-one RT Mix with gDNA Remover (GenStar, Beijing, China). For miRNA, RNA samples were reverse transcribed into cDNA using a StarScript III miRNA RT Kit with stem-loop primers (GenStar, Beijing, China). The qPCR reaction mixture for both mRNA and miRNA was 10 μL in total volume, including 5 μL of SYBR Green I mix (GenStar, Beijing China), 4 μL of DEPC water, and 1 μL of cDNA synthesized from mRNA (10 ng) and miRNA (20 ng), as well as 0.5 μL each of the specific forward and reverse primers (10 µM) for mRNA or miRNA detection. Real-time fluorescence quantification was performed using the 2 × RealStar Fast SYBR qPCR Mix (Low ROX) (GenStar, Beijing, China) on the ABI QuantStudio 7 real-time PCR instrument (Thermo Fisher Scientific Inc., Waltham, MA, USA), with three technical replicates for each sample. *GAPDH* was used as the reference gene for mRNA, while U6 snRNA was used for miRNA. The cycling conditions were 95 °C for 15 s, followed by annealing at 60 °C for 1 min, for a total of 40 cycles, with fluorescence signals collected at the extension phase of each cycle. The primers used in this study are listed in Appendix A. The relative expression levels of each type of RNA were calculated using the 2^−ΔΔCt^ method [33], and statistical analysis and graphs were generated using GraphPad Prism 9.50 software (GraphPad Software Inc., San Diego, CA, USA).

## 3. Results

### 3.1. Summary of mRNA Sequencing Data

The transcriptomes of six individuals, including three gray-feathered and three white-feathered Shitou geese, were sequenced to generate approximately 262.7 million (43,781,091 on average) raw reads. After the removal of low-quality reads and adapter sequences, 260.9 million (43,483,615 on average) clean reads were obtained. The GC content and Q30 percentages per library were more than 50% and 93%, respectively. The proportion of clean reads mapped to the *Anser cygnoides* reference genome in each library exceeded 80%, and the proportion of uniquely mapped reads ranged from 75.88% to 79.43% (Table 1).

### 3.2. Differentially Expressed Genes between Gray and White Feather Follicles

After comparing the gene expression levels between gray and white feather follicles, a total of 329 significantly differentially expressed genes (DEGs) were obtained, of which 132 genes were significantly up-regulated while 197 genes were significantly down-regulated in white feather follicles. The expression of twelve well-known pigmentation genes (*TYR*, *TYRP1*, *EDNRB2*, *MLANA*, *SOX10*, *SLC45A2*, *GPR143*, *TRPM1*, *OCA2*, *ASIP*, *KIT*, *SLC24A5*) were significantly down-regulated in white feather follicles (FDR < 0.05, |log2FC| > 1, Figure 2, Appendix A).

### 3.3. Differentially Expressed miRNAs and Their Target Genes in the Gray and White Feather Follicles

Six small RNA libraries were constructed and then sequenced separately using Illumina Novaseq6000. After discarding low-quality reads and masking adaptor sequences, a total of 74,494,198 clean tags were obtained, with the vast majority being 22 nt in length, which is consistent with the typical size of miRNAs. Following the removal of other classes of small RNAs, such as rRNA, tRNA, snRNA, snoRNA, etc., the remaining clean reads were aligned to the reference genome, and the ratio of matched tags from six samples ranged from 70% to 77% (Figure 3A, Appendix A). The search in the miRBase database identified a total of 686 known miRNAs, while mirdeep2 (version 0.1.3) software predicted an additional set of 325 novel candidate miRNAs across six libraries (Appendix A).

A total of 46 DEMs were identified in gray and white feather follicles, of which 15 were significantly up-regulated and 31 were significantly down-regulated in white feathers. *miR-451-x* and *miR-204-x* were identified as the most highly expressed up-regulated and down-regulated miRNAs in white feathers, respectively (*p*-value < 0.05, |log2FC| > 1, Figure 3B, Appendix A). Given the large number of target genes for each miRNA, we focused on target genes that were differentially expressed in feather follicles. Therefore, we identified 205 target DEGs in the gray and white feather follicles, including the following eight aforementioned pigmentation-related genes: *TYRP1*, *EDNRB2*, *MLANA*, *SOX10*, *GPR143*, *TRPM1*, *KIT*, and *SLC24A5* (Appendix A).

### 3.4. Analysis of the Constructed miRNA and mRNA Network

To investigate the potential regulatory network underlying the formation of white feathers in the Shitou goose, we constructed an association network of DEMs and DEGs based on their expression level correlation. Considering the negative regulation mechanism of miRNAs on its target genes, we focused on miRNAs that were negatively correlated with the expression of pigmentation-related genes. Five miRNAs including *novel-m0086-5p*, *miR-489-y*, *miR-223-x*, *miR-7565-z*, and *miR-3535-z* exhibited a significant negative correlation with pigment genes such as *TYRP1*, *EDNRB2*, *MLANA*, *SOX10*, *TRPM1*, and *KIT* (Figure 4, Appendix A).

### 3.5. Validation of RNA and miRNA Expression Results by qRT-PCR

To validate the accuracy and reliability of RNA sequencing, the expression levels of four key pigment genes (*TYR*, *EDNRB2*, *SOX10*, *KIT*) were measured using qPCR of feather follicle tissues from Shitou geese with gray and white plumage. The results showed that the expression of *TYR*, *EDNRB2*, *SOX10*, and *KIT* were significantly lower in the white feather follicles than in the gray feather follicles in Shitou geese (*p* < 0.05), which is consistent with the sequencing results of the transcriptome, indicating the reliability of the sequencing results (Figure 5A). Similarly, the expression levels of four miRNAs (*miR-451-x*, *miR-144-y*, *miR-204-x*, *miR-196-x*) were verified by qPCR, exhibiting consistent results with the sequencing data (Figure 5B).

## 4. Discussion

In avian species, variations in plumage coloration primarily stem from alterations in the synthesis, distribution, and deposition of melanin specifically within feather follicles, which are the sole sites of plumage melanin production. Within these follicles, mature melanocytes undertake the crucial task of synthesizing melanosomes, which are essential for melanogenesis. Subsequently, the newly formed melanin is transferred to the keratinocytes of the feathers, conferring them with their distinctive coloration [34,35,36]. In this study, we identified a number of DEGs that are well-known for pigmentation, encompassing *TYR*, *TYRP1*, *EDNRB2*, *MLANA*, *SOX10*, *SLC45A2*, *GPR143*, *TRPM1*, *OCA2*, *ASIP*, *KIT*, and *SLC24A5*, which were significantly down-regulated in white feather follicles of Shitou goose by RNA sequencing. These genes play crucial roles in various aspects of melanin synthesis, melanocyte development, and pigment transport pathways.

Endothelin receptor type B-like (*EDNRB2*), SRY-box transcription factor 10 (*SOX10*), and Kit Proto-Oncogene Receptor Tyrosine Kinase (*KIT*) play pivotal roles in melanocyte migration, proliferation, and differentiation, as well as the development of neural crest cells, which are precursors to melanocytes [37,38,39]. *EDNRB2* is crucial for melanocyte formation and migration, and mutations in this gene can lead to a range of feather color phenotypes, from white spots to complete whiteness in avian species [5,9,10,40,41,42,43,44]. Similarly, mutations in *SOX10* have been implicated in the development of different plumage colors in chickens, including light yellow and dark brown hues [45,46]. The *KIT* gene, a member of the tyrosine kinase growth factor receptor family, encodes proteins that regulate the proliferation and growth of melanocytes [47]. Mutations in *KIT* disrupt the normal migration of melanocyte precursors and melanocyte formation, resulting in the absence of melanocytes and melanin granules in hair follicles. This, in turn, leads to a white coloration phenotype [12,39,48,49,50,51].

Tyrosinase (*TYR*) and Tyrosinase-related protein 1 (*TYRP1*) are fundamental in melanin synthesis. Mutations in *TYR* can lead to albinism, a condition characterized by the absence of pigmentation in skin, fur, and feathers because of the inability to produce melanin [52]. Similarly, mutations in *TYRP1* can affect the conversion of 5,6-dihydroxyindole, a precursor of brownish melanin, into blackish eumelanin, leading to altered pigmentation phenotypes [53,54]. In chickens, for instance, the insertion of a retroviral sequence into intron 4 of the *TYR* gene has been identified as the cause of a recessive white feather color trait [55]. On the other hand, mutations in the *TYRP1* gene have been associated with brown coloration phenotypes in various animals, including pigs, goats, dogs, and chickens [56,57,58,59]. Additionally, Agouti Signaling Protein (*ASIP*) plays a regulatory role in melanin synthesis by modulating the balance between eumelanin and pheomelanin production [2]. Variants at the *ASIP* locus have been found to contribute to yellow mutation in Japanese quail, coat color darkening in Nellore cattle, and the bay coat color in horses [60,61,62].

Melan-A (*MLANA*), Transient Receptor Potential Cation Channel Subfamily M Member 1 (*TRPM1*), Solute Carrier Family 45 Member 2 (*SLC45A2*), Solute Carrier Family 24 Member 5 (*SLC24A5*), G Protein-coupled Receptor 143 (*GPR143*), and Oculocutaneous Albinism II (*OCA2*) are all genes that play crucial roles in melanosome function and melanin transport. Variations in the expression levels of these genes can significantly influence melanin distribution and the intensity of skin coloration [63]. For instance, *MLANA* is involved in melanosome biogenesis and maturation, linked to white plumage in greylag geese and almond color in domestic pigeons [64]. *TRPM1* regulates calcium influx into melanocytes, affecting melanosome movement and melanin synthesis [65]. *SLC45A2* and *SLC24A5* are involved in melanin synthesis and transport, and mutations in *SLC45A2* affect feather coloration in chicken and quail, while variants in *SLC24A5* are associated with “tiger-eye” iris pigmentation in horses [66,67]. *GPR143* is associated with melanosome dispersion and pigmentation patterns, and mutations in *GPR143* cause X-linked Ocular Albinism Type 1 in humans [68]. *OCA2* is a key regulator of melanin synthesis, affecting both eumelanin and pheomelanin production, with a splice site variant in the canine *OCA2* gene associated with the oculocutaneous albinism phenotype in dogs [69].

Generally, any variations in the development of melanocytes and the synthesis of melanin have the potential to disrupt the process of plumage color formation, ultimately resulting in white plumage. Our observations showed that the mature melanocyte markers *TYR* and *TYRP1* exhibit minimal expression in the white feather samples, suggesting an abnormal development of melanocytes. Given the established roles of *EDNRB2* and *KIT* in regulating the migration, proliferation, and differentiation of melanocytes, it is conceivable that their involvement could potentially contribute to the failure of melanocyte formation or migration [37,39]. Such failures would effectively preclude the melanocytes from synthesizing the necessary pigment, ultimately leading to the formation of white feathers. Furthermore, our analysis revealed that the expression levels of *EDNRB2* and *KIT* in white feather samples were significantly down-regulated compared to their expression in gray feather samples. Notably, the *EDNRB2* gene exhibited virtually no expression in the white feather samples, suggesting a critical role in the absence of pigmentation.

Taken together, these findings support the hypothesis that disturbances in the melanocyte development and melanin synthesis pathways, specifically involving *EDNRB2* and *KIT*, underlie the formation of white feathers in Shitou geese. The reduced expression of these genes may hinder melanocyte migration, proliferation, and differentiation, ultimately leading to the inability to produce pigmented feathers. Additional studies are needed to fully elucidate the complex interactions among these genes and their impact on feather coloration in Shitou geese. In previous studies, a 14 bp insertion in the *EDNRB2* gene and an 18 bp deletion in the *KIT* gene were associated with white feathers in Chinese domestic goose breeds [5,9,10,12]. Investigating whether Shitou geese also possess these mutations and dissecting the molecular mechanism of the formation of white plumage in Shitou geese will be the primary focus of our future research.

miRNAs are crucial post-transcriptional regulators and transcriptional gene network modulators that significantly influence animal skin and hair coloration [70,71,72,73]. To further explore the regulatory networks and potential roles of miRNAs in the formation of white plumage in the Shitou goose, we conducted deep sequencing and bioinformatics analysis to identify DEMs in gray and white feather follicle tissues. Our comprehensive analysis revealed the presence of 46 DEMs across the six libraries, with 15 significantly up-regulated and 31 significantly down-regulated in white feathers. Given that miRNAs typically function to suppress the expression of their target genes, the up-regulation of miRNAs in white feathers suggests their role in suppressing the expression of pigmentation-related genes. Notably, *miR-451-x*, *miR-144-y*, and *miR-2188-x* emerged as the top three most abundantly expressed and up-regulated miRNAs in white feathers. Currently, there is limited direct evidence linking these specific miRNAs to feather coloration. *miR-451* has primarily been studied in the context of its involvement in cancers [74,75,76], indicating its diverse functions beyond pigmentation. *Dre-miR-2188*, on the other hand, has been implicated in targeting *Nrp2a* and regulating proper intersegmental vessel development in zebrafish embryos [77], suggesting a role in embryonic vascular development. *miR-144-y* has been associated with developmental processes in the skin and feather follicles of Zhedong white geese, offering valuable insights into the potential function of *miR-144-y* in feather development [78]. On the other hand, *miR-144-3p* has been identified as a component of *KIT*-related regulatory networks in human Gastrointestinal Stromal Tumors (GISTs), suggesting its potential role in regulating the expression of the *KIT* gene [79]. However, the specific role of *miR-144-y* in feather coloration remains largely unexplored. 

Additionally, our analysis revealed that the expression of *novel-m0086-5p*, *miR-489-y*, *miR-223-x*, *miR-7565-z*, and *miR-3535-z* exhibits a significant negative correlation with the expression of pigmentation-related genes such as *TYRP1*, *EDNRB2*, *MLANA*, *SOX10*, *TRPM1*, and *KIT*. This intriguing finding suggests that these miRNAs may regulate feather pigmentation through indirect mechanisms. Given that miRNAs typically function by suppressing the expression of their target genes, the negative correlation observed between these miRNAs and pigmentation genes implies that they may be involved in regulating the expression of genes that are upstream or downstream of the pigmentation pathway [14]. Alternatively, they could potentially regulate the expression of transcription factors or cofactors that modulate the activity of pigmentation genes.

The identification of these miRNAs as potential regulators of feather pigmentation represents a significant step toward understanding the complex regulatory networks underlying this trait. However, we must acknowledge that the small sample size in this study, consisting of only three biological replicates for each type of feather, represents a limitation that may impact the statistical power and generalizability of our results. As previously discussed by Kok et al. (2017), small sample sizes in high-throughput screens can be a pitfall for identifying reliable biomarkers, which may cause small sample size errors [80]. Although our findings provide preliminary insights into the miRNA profiles of different feather color types, future studies with larger sample sizes are needed to validate and strengthen the confidence of the results. Moreover, future studies aimed at elucidating the specific targets and functional mechanisms of these miRNAs will provide deeper insights into the regulatory processes governing feather coloration in Shitou geese. Such insights could potentially lead to the development of novel strategies for manipulating feather color through genetic engineering, which could have important implications for the poultry industry and beyond.

## 5. Conclusions

In this study, transcriptome sequencing analysis revealed significant down-regulation of twelve well-known pigmentation genes (*TYR*, *TYRP1*, *EDNRB2*, *MLANA*, *SOX10*, *SLC45A2*, *GPR143*, *TRPM1*, *OCA2*, *ASIP*, *KIT*, *SLC24A5*) in white feather follicles. Of these, the *EDNRB2* and *KIT* genes stood out as the most likely candidates for white feather formation in Shitou geese. Furthermore, our analysis revealed that *miR-144-y* may play crucial roles in the regulation of feather pigmentation, and the expression of *novel-m0086-5p*, *miR-489-y*, *miR-223-x*, *miR-7565-z*, and *miR-3535-z* exhibits a significant negative correlation with the expression of pigmentation-related genes.

## Figures and Tables

**Figure 1 animals-14-01508-f001:**
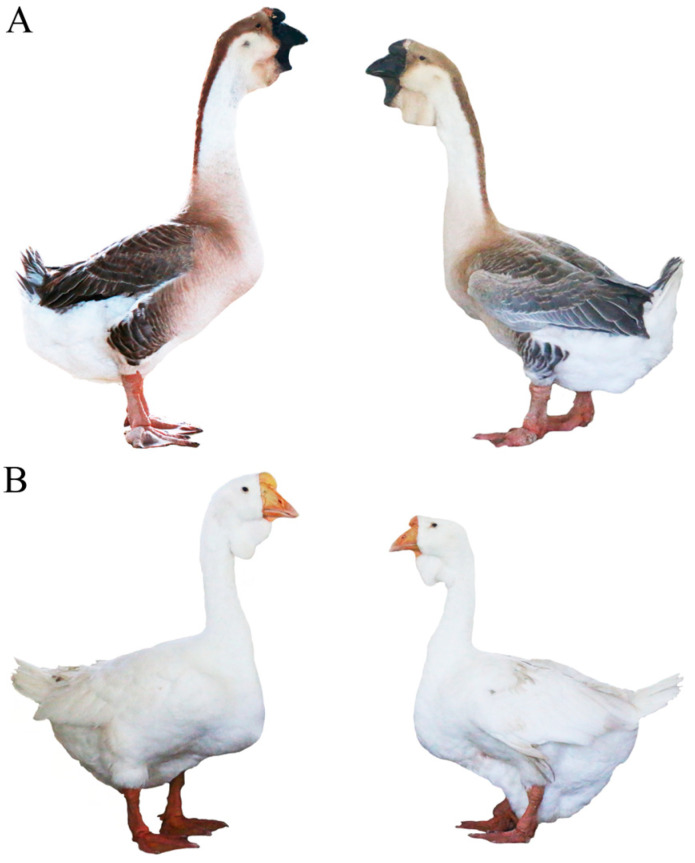
Phenotypic comparison between gray and white Shitou geese. (**A**) Male (**left**) and female (**right**) gray-feathered Shitou geese. (**B**) Male (**left**) and female (**right**) white-feathered Shitou geese.

**Figure 2 animals-14-01508-f002:**
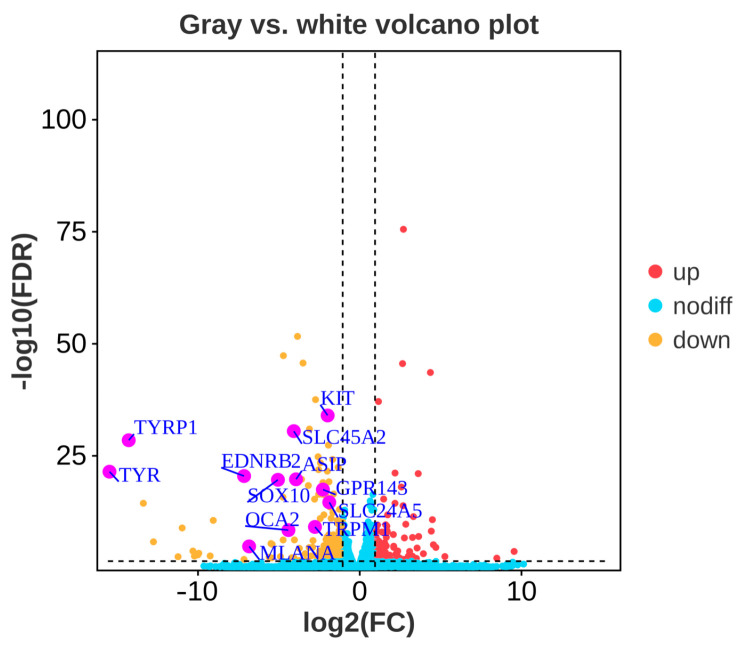
Volcano plot of differentially expressed genes between gray and white feather follicles in Shitou geese. Tyrosinase (*TYR*), Tyrosinase-related protein 1 (*TYRP1*), Endothelin receptor type B-like (*EDNRB2*), Melan-A (*MLANA*), SRY-box transcription factor 10 (SOX10), Solute Carrier Family 45 Member 2 (*SLC45A2*), G Protein-coupled Receptor 143 (*GPR143*), Transient Receptor Potential Cation Channel Subfamily M Member 1 (*TRPM1*), Oculocutaneous Albinism II (OCA2), Agouti Signaling Protein (*ASIP*), Kit Proto-Oncogene Receptor Tyrosine Kinase (*KIT*), Solute Carrier Family 24 Member 5 (*SLC24A5*).

**Figure 3 animals-14-01508-f003:**
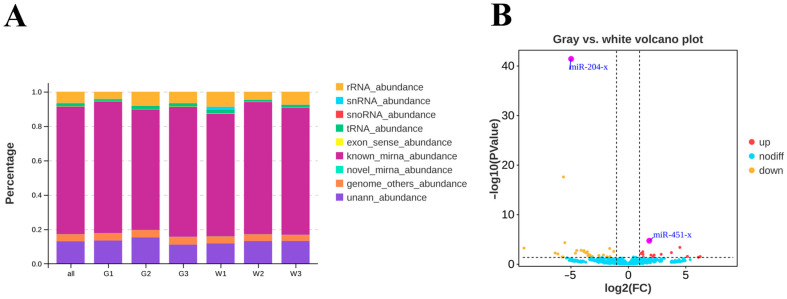
Analysis of differentially expressed miRNAs in gray and white feather follicles of Shitou geese. (**A**) The ratio of different types of small RNA in each sample; all: all samples; G1: Gray feather1; G2: Gray feather2; G3: Gray feather3; W1: White feather1; W2: White feather2; W3: White feather3. (**B**) Volcano map of differentially expressed miRNAs from gray and white feathers.

**Figure 4 animals-14-01508-f004:**
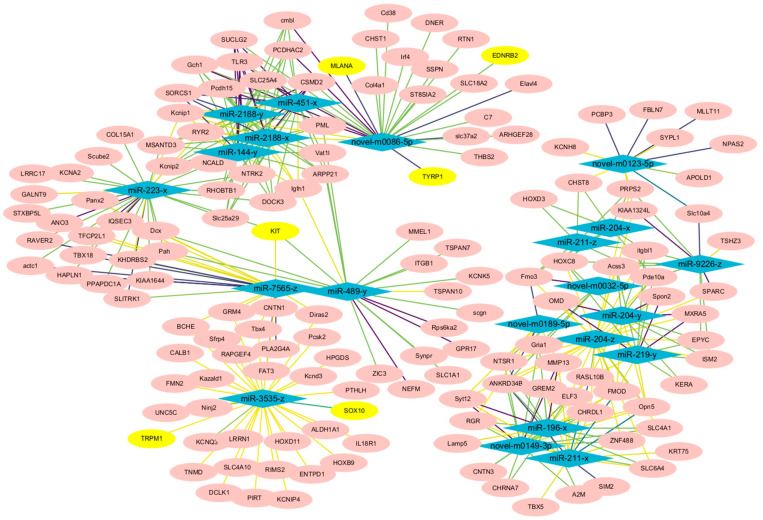
Construction of the miRNA-mRNA interaction network based on expression level correlations. The pink ellipses represent differentially expressed genes, the yellow ellipses represent pigment genes, and the blue diamonds represent differentially expressed miRNAs. The darker the color of the connecting lines, the greater the correlation coefficient.

**Figure 5 animals-14-01508-f005:**
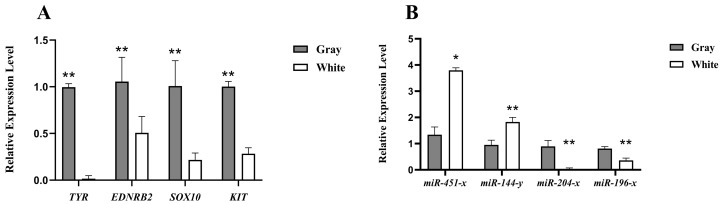
Verification of the sequencing results using qRT-PCR analysis. (**A**) Relative expression of Tyrosinase (*TYR*), Endothelin receptor type B-like (*EDNRB2*), SRY-box transcription factor 10 (SOX10) and Kit Proto-Oncogene Receptor Tyrosine Kinase (*KIT*) gene in gray and white feather follicles of Shitou geese. (**B**) Relative expression of *miR-451-x*, *miR-144-y*, *miR-204-x*, and *miR-196-x* in gray and white feather follicles of Shitou geese. * Indicates a significant difference (*p* < 0.05). ** Indicates an extremely significant difference (*p* < 0.01).

**Table 1 animals-14-01508-t001:** Statistical analysis of transcriptome sequencing data.

Sample	Raw Reads	Clean Reads	Clean Bases (bp)	GC_Content (%)	Q30_Value (%)	Total_Mapped (%)	Unique_Mapped (%)
G1	43,590,714	43,270,040	6,450,918,824	50.18	93.70	82.18	78.74
G2	445,30,290	44,220,580	6,591,098,695	50.26	94.17	83.01	79.02
G3	43,000,870	42,707,118	6,369,209,432	50.06	93.58	82.52	79.43
W1	43,590,238	43,295,454	6,455,337,938	51.32	93.87	80.89	76.74
W2	44,149,770	43,872,124	6,539,210,760	51.51	94.25	80.70	75.88
W3	43,824,664	43,536,374	6,501,179,292	50.86	93.49	81.12	77.50

G1: Gray feather1; G2: Gray feather2; G3: Gray feather3; W1: White feather1; W2: White feather2; W3: White feather3.

## Data Availability

The data presented in this study are available in the Appendix A.

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
