# Peer review of "Identification of Differentially Expressed Genes and microRNAs in the Gray and White Feather Follicles of Shitou Geese"

_animals, 2024, doi:10.3390/ani14101508_

Round 1

Reviewer 1 Report

Comments and Suggestions for Authors

1.Please give a brief introduction to the development process of goose feather color. Are there any differences in feathering color between male and female geese?

2.In Figure 1, there is not a picture of female goose. Please add one.

3.The sex, age and specific sampling location of the samples are not clearly described.

4.Line49-50, .Therefore, white feathers are a common phenomenon in commercial breeds”. I do not think so. At least some commercial layer breeds show colored feathers.

5.  Line 116, please add the information of read length and SE or PE mode.

6.Line128, “fastq software”, what is it? Please give its name and version.

7.Line 133 and 145, absolute fold change > 1, do you mean |log2FC| > 1? please revise it.

8.   In the discussion section, references should be added to every sentence involving the research results of others.

9.  Line 358-360,“Of these, EDNRB and KIT genes stood out as potentially crucial regulators of white feather formation in Shitou geese.”, This is a little bit arbitrary.

Author Response

Dear reviewer,

We are grateful to your comments on our manuscript entitled “Identification of differentially expressed genes and microRNAs in the gray and white feather follicles of Shitou geese” (animals-3001104). These comments help us make an improvement of this paper. We have made revisions according to these comments. In addition, we have also corrected the spelling and punctuation errors in the article. All revisions are indicated in red in the manuscript. The revised figures and supplementary tables have been resubmitted in the attachment. We sincerely hope that these revisions would satisfy you. The point-to-point responses to the comments are shown as follows. Please do not hesitate to contact me if you have any other questions or comments.

It would be greatly appreciated if our revised manuscript could be considered for publication in Animals.

Best regards,

Dr. Zhongping Wu

-----------------------------

The following is a point-to-point response to your comments.

Comments:

1.Please give a brief introduction to the development process of goose feather color. Are there any differences in feathering color between male and female geese? 

Response: Thanks for this comment. We have added a sentence in lines 58-61 to introduce the development process of goose feather color. Unlike chicken and duck, geese including the greylag and swan geese, typically do not exhibit sexual dimorphism in adults. However, it has been reported that sexual dimorphism occasionally occurs in greylag goslings.

2.In Figure 1, there is not a picture of female goose. Please add one. 

Response: Changed.

3.The sex, age and specific sampling location of the samples are not clearly described. 

Response: Added. Please see lines 115 and 117.

4.Line49-50, “.Therefore, white feathers are a common phenomenon in commercial breeds”. I do not think so. At least some commercial layer breeds show colored feathers. 

Response: Thanks to this comment. The previous description is indeed not rigorous enough, we have revised it. Please see lines 52-53.

5.Line 116, please add the information of read length and SE or PE mode. 

Response: Added. Please see lines 134.

6.Line128, “fastq software”, what is it? Please give its name and version. 

Response: Sorry. That's a spelling mistake. It should be “fastp software”. We have revised and give its version. Please see lines 149.

7.Line 133 and 145, “absolute fold change > 1”, do you mean “|log2FC| > 1”?

Response: Yes. We have revised. Please see lines 156 and 168.

8.In the discussion section, references should be added to every sentence involving the research results of others. 

Response: We are grateful to this suggestion. We have added additional references in the revised manuscript.

9.Line 358-360,“Of these, EDNRB and KIT genes stood out as potentially crucial regulators of white feather formation in Shitou geese.”,This is a little bit arbitrary. 

Response: We are grateful to this comment. That's indeed not rigorous enough. We have revised. Please see lines 424-425.

Reviewer 2 Report

Comments and Suggestions for Authors

Dear authors:

The main objective of this study was to evaluate the precise genetic mechanism of plumage color (grey or white) in Shitou geese through transcriptome sequencing and microRNA techniques. This is important because of down feather market and meat producers preferences and because of genetic correlations that can benefit from these phenotypes.

The paper is well-written, original and interesting. I have only one suggestion: the sample size is very small and it arguably can have an impact in the statistical tests of hypothesis used in the study (two-way ANOVA, among others).

Although the cost of these types of study are too elevated to generate a large sample size it is important to at least make a paragraph to comment the limitations of the study concerning statistical power. See the reference: Kok MGM, de Ronde MWJ, Moerland PD, Ruijter JM, Creemers EE, Pinto-Sietsma SJ. Small sample sizes in high-throughput miRNA screens: A common pitfall for the identification of miRNA biomarkers. Biomol Detect Quantif. 2017 Dec 18;15:1-5. doi: 10.1016/j.bdq.2017.11.002. PMID: 29276692; PMCID: PMC5737945.

Author Response

Dear reviewer,

We are grateful to your comments on our manuscript entitled “Identification of differentially expressed genes and microRNAs in the gray and white feather follicles of Shitou geese” (animals-3001104). These comments help us make an improvement of this paper. We have made revisions according to these comments. In addition, we have also corrected the spelling and punctuation errors in the article. All revisions are indicated in red in the manuscript. The revised figures and supplementary tables have been resubmitted in the attachment. We sincerely hope that these revisions would satisfy you. The point-to-point responses to the comments are shown as follows. Please do not hesitate to contact me if you have any other questions or comments.

It would be greatly appreciated if our revised manuscript could be considered for publication in Animals.

Best regards,

Dr. Zhongping Wu

-----------------------------

The following is a point-to-point response to your comments.

Comments:

1.The main objective of this study was to evaluate the precise genetic mechanism of plumage color (grey or white) in Shitou geese through transcriptome sequencing and microRNA techniques. This is important because of down feather market and meat producers preferences and because of genetic correlations that can benefit from these phenotypes.

Response: We greatly appreciate this positive general comment.

2.The paper is well-written, original and interesting. I have only one suggestion: the sample size is very small and it arguably can have an impact in the statistical tests of hypothesis used in the study (two-way ANOVA, among others). 

Response: Thanks to this comment. We have answered this question below.

3.Although the cost of these types of study are too elevated to generate a large sample size it is important to at least make a paragraph to comment the limitations of the study concerning statistical power. See the reference: Kok MGM, de Ronde MWJ, Moerland PD, Ruijter JM, Creemers EE, Pinto-Sietsma SJ. Small sample sizes in high-throughput miRNA screens: A common pitfall for the identification of miRNA biomarkers. Biomol Detect Quantif. 2017 Dec 18;15:1-5. doi: 10.1016/j.bdq.2017.11.002. PMID: 29276692; PMCID: PMC5737945. 

Response: We agree with this comment. Indeed, due to the cost factors, we only did three biological replicates for each type of feathers in our study. We must acknowledge that there might cause small sample error in our results. Hence, we added additional sentences to comment the limitations of this study in the revised manuscript. Please see lines 407-414.

Reviewer 3 Report

Comments and Suggestions for Authors

Guo et al. studied the genetic basis of feather pigmentation and lack of pigment (white phenotype) in feather follicles of a certain Asian goose breed, the Shitou goose. The authors identified several known pigment related genes to be downregulated in white feather follicles and postulated EDNRB and KIT genes to be strong candidates behind the white feather plumage. The authors also identified several miRNAs to be up- or downregulated in white feather follicles. Three miRNA genes were found to be most upregulated in white feather follicles, but lacked direct link to pigment related genes. Several miRNAs were identified to be correlated with lack of expression of pigment related genes, hinting that miRNAs could play a role in goose feather pigmentation regulation.

Goose feather coloration has been studied previously to some extend, particularly for the breeds descending from the swan goose, but to my knowledge, the role of miRNAs related to the plumage phenotypes is much less explored. This study provides more insights into the basis of white plumage colour of Asian goose breeds in line with previously published studies. However, it would have been interesting if the authors would have genotyped the geese for the 14-bp insertion in the exon 3 of EDNRB2 gene, known to underly white plumage in other Asian goose breeds. Mainly the text was well-written, but there were some spelling errors here and there. The methods should be written in more detail to be replicable. The figures were too small in the present form. Below, I list improvement suggestions line-by-line:

Line 16: Change “A” to “a” before “transcriptome” and change “analysis” to “analysis”

Line 21: Remove extra spaces between “gray” and “plumage”

Line 40 Keywords: Use different keywords than in the title, add Anser cygnoides

Line 57-59: Geese have been domesticated from two wild species, the Asian breeds (excluding Yili goose) from the swan goose (Anser cygnoides), known as Chinese domestic goose, and European breeds from the greylag goose (Anser anser), known as European domestic goose. Please add this information here

Line 62: Please add the scientific name after Shitou geese so that is evident that the Shitou goose originates from the swan goose, so “Shitou geese (A. cygnoides)…”

Line 68-73: Also the 14-bp insertion is reported in Yang et al. 2022 Using comparative genomics to detect mutations regulating plumage variations in graylag (A. anser) and swan geese (A. cygnoides). Gene 834:146612. Please add as a reference

Line 107: Change “hair” to “feather”

Line 114: Please describe the mRNA selection, library preparation and transcriptome sequencing in more detail. How you made cDNA etc. Describe the methods so that someone else could repeat the procedures.

Line 119-124: Describe the methods here as well in more detail. Describe the enrichment process as well the RT-PCR and library preparation so that these could be repeated by someone else

Line139-140: Why miRNAs mapped to introns or exons were removed? Non-coding RNAs can be found from introns or also on exons (e.g. some non-coding RNAs are in different orientation to the gene within exon).

Line 154: Reference to Cytoscape is missing, please add

Line 157-168: It is unclear which mRNA was used. Only from the supplementary table, it can be seen that this quantitative RT-PCR was used to validate results for certain genes. Please describe for what purpose this validation process was and which were the validated genes. Please describe the PCR protocols. In Supplementary Table S1, for protein-coding genes there is F and R primer but for miRNAs there is additional RT primer. Please explain this

Line 168: Reference to GraphPad Prism software is missing, please add

Line 182-184: Remove figure 2A. You say the same information in the text as in the Fig.2A, so the figure does not add any information to the text.

Line 188: Make Fig. 2B bigger as the text is unreadable. Remove the A part of the figure and this frees more space to make the B part bigger and increase the text size within the figure.

Line 189-191: As Figure legends should be readable without the main text, please open the abbreviation DEG in the figure legend.

Line 199-201: Figure 3C does not provide additional information to the text, remove Fig. 3A.

Line 207: Correct spelling of “therefore”

Line 211: The figures and the text within the figures is too small. Please make figures and text bigger.

Line 212: As Figure legends should be readable without the main text, please open the abbreviation DEM in the figure legend.

Line 224: Please make the circles and the text bigger as it is impossible to read the gene names from the figure

Line 225-228: Please also explain what the yellow colour stands for

Line 242: Font for the miRNA gene names seems bigger than in the other text, please correct

Line 260-262: In avian species, usually the EDNRB2 gene is the one with mutations associated plumage color, also in the references you have cited are about the EDBRB2 gene, not EDNRB (these are different paralogous genes). Also in online mendelian inheritance of animals (OMIA), I havent’t found any avian species with mutations reported in EDNBR, all mutations have been in EDNRB2 gene. Could you check that the gene meant here is really EDNRB

Line 287: MLANA has been shown to be associated with white plumage in greylag geese (Yang et al. 2022 Using comparative genomics to detect mutations regulating plumage variations in graylag (A. anser) and swan geese (A. cygnoides). Gene 834:146612) and to almond color in domestic pidgeons (Bruders et al. (2020) A copy number variant is associated with a spectrum of pigmentation patterns in the rock pigeon (Columba livia). PLoS Genet 16(5): e1008274.

Line 289: SLC45A2 mutation known to affect feather coloration in chicken and quail (Gunnarsson 2007 Mutations in SLC45A2 Cause Plumage Color Variation in Chicken and Japanese Quail. Genetics 175: 867–877)

Line 307: Could be pointed out here that in future it would be interesting to genotype the Shitou geese for the 14-bp insertion in the EDNRB2 gene to see if the same mutation identified in several other Chinese domestic goose breeds is behind the white plumage also in Shitou geese

Supplementary materials. Please provide better table titles which explain what is in each column of the tables or change column titles to be more informative. For example ST-G4_count  is not informative, what is ST and G4. Or G4_TPM, what is G4 and what is TPM? avoid abbreviation without explaining them.

Comments on the Quality of English Language

Some spelling errors were detected, please go trough the text carefully

Author Response

Dear reviewer,

We are grateful to your comments on our manuscript entitled “Identification of differentially expressed genes and microRNAs in the gray and white feather follicles of Shitou geese” (animals-3001104). These comments help us make an improvement of this paper. We have made revisions according to these comments. In addition, we have also corrected the spelling and punctuation errors in the article. All revisions are indicated in red in the manuscript. The revised figures and supplementary tables have been resubmitted in the attachment. We sincerely hope that these revisions would satisfy you. The point-to-point responses to the comments are shown as follows. Please do not hesitate to contact me if you have any other questions or comments.

It would be greatly appreciated if our revised manuscript could be considered for publication in Animals.

Best regards,

Dr. Zhongping Wu

-----------------------------

The following is a point-to-point response to your comments.

Comments:

1.Guo et al. studied the genetic basis of feather pigmentation and lack of pigment (white phenotype) in feather follicles of a certain Asian goose breed, the Shitou goose. The authors identified several known pigment related genes to be downregulated in white feather follicles and postulated EDNRB and KIT genes to be strong candidates behind the white feather plumage. The authors also identified several miRNAs to be up- or downregulated in white feather follicles. Three miRNA genes were found to be most upregulated in white feather follicles, but lacked direct link to pigment related genes. Several miRNAs were identified to be correlated with lack of expression of pigment related genes, hinting that miRNAs could play a role in goose feather pigmentation regulation.

Response: We are grateful to this positive comment.

2.Goose feather coloration has been studied previously to some extend, particularly for the breeds descending from the swan goose, but to my knowledge, the role of miRNAs related to the plumage phenotypes is much less explored. This study provides more insights into the basis of white plumage colour of Asian goose breeds in line with previously published studies. However, it would have been interesting if the authors would have genotyped the geese for the 14-bp insertion in the exon 3 of EDNRB2 gene, known to underly white plumage in other Asian goose breeds. Mainly the text was well-written, but there were some spelling errors here and there. The methods should be written in more detail to be replicable. The figures were too small in the present form. Below, I list improvement suggestions line-by-line:

Response: We are grateful to these suggestions. Based on this study, our future focus will be on investigating the relationship between the variation of identified pigment genes and their impact on the feather colors of Shitou geese. In particular, we will  explore whether the 14 bp insertion in exon 3 of the EDNRB2 gene is contribute to the white feathers in Shitou geese. Then, we will further dissect the molecular mechanism of the formation of white feathers in Shitou geese. We sincerely appreciate your thorough review and valuable suggestions, and we will make revisions accordingly.

3.Line 16: Change “A” to “a” before “transcriptome” and change “analysis” to “analysis”

Response: Changed. Please see lines 16.

4.Line 21: Remove extra spaces between “gray” and “plumage”

Response: Changed. Please see lines 22.

5.Line 40 Keywords: Use different keywords than in the title, add Anser cygnoides

Response: Changed. Please see lines 41-42.

6.Line 57-59: Geese have been domesticated from two wild species, the Asian breeds (excluding Yili goose) from the swan goose (Anser cygnoides), known as Chinese domestic goose, and European breeds from the greylag goose (Anser anser), known as European domestic goose. Please add this information here

Response: Added. Please see lines 60-62.

7.Line 62: Please add the scientific name after Shitou geese so that is evident that the Shitou goose originates from the swan goose, so “Shitou geese (A. cygnoides)…”

Response: Changed. Please see lines 68.

8.Line 68-73: Also the 14-bp insertion is reported in Yang et al. 2022 Using comparative genomics to detect mutations regulating plumage variations in graylag (A. anser) and swan geese (A. cygnoides). Gene 834:146612. Please add as a reference

Response: Added. Please see lines 81.

9.Line 107: Change “hair” to “feather”

Response: Changed. Please see lines 105.

10.Line 114: Please describe the mRNA selection, library preparation and transcriptome sequencing in more detail. How you made cDNA etc. Describe the methods so that someone else could repeat the procedures.

Response: Changed. Please see lines 124-134.

11.Line 119-124: Describe the methods here as well in more detail. Describe the enrichment process as well the RT-PCR and library preparation so that these could be repeated by someone else

Response:  Changed. Please see lines 137-139.

12.Line139-140: Why miRNAs mapped to introns or exons were removed? Non-coding RNAs can be found from introns or also on exons (e.g. some non-coding RNAs are in different orientation to the gene within exon).

Response: Because those mapped to exons or introns might be fragments from mRNA degradation, so these tags were removed.

13.Line 154: Reference to Cytoscape is missing, please add

Response: Added. Please see lines 179.

14.Line 157-168: It is unclear which mRNA was used. Only from the supplementary table, it can be seen that this quantitative RT-PCR was used to validate results for certain genes. Please describe for what purpose this validation process was and which were the validated genes. Please describe the PCR protocols. In Supplementary Table S1, for protein-coding genes there is F and R primer but for miRNAs there is additional RT primer. Please explain this

Response: We are grateful to these suggestions. We added additional sentences to delineate the purpose and the genes and miRNAs chosen for validation. Please see lines 181-190. The PCR protocols were described in lines 193-197. When amplifying miRNAs, an additional stem-loop reverse transcription (RT) primer is often utilized alongside forward and reverse PCR primers. This necessity arises because miRNAs, being small RNA molecules approximately 22 nucleotides long, do not lend themselves to direct PCR amplification as do longer DNA sequences.

15.Line 168: Reference to GraphPad Prism software is missing, please add

Response: Added. Please see lines 207.

  1. Line 182-184: Remove figure 2A. You say the same information in the text as in the Fig.2A, so the figure does not add any information to the text.

Response: Changed.

  1. Line 188: Make Fig. 2B bigger as the text is unreadable. Remove the A part of the figure and this frees more space to make the B part bigger and increase the text size within the figure.

Response: Changed.

18.Line 189-191: As Figure legends should be readable without the main text, please open the abbreviation DEG in the figure legend.

Response: Changed. Please see lines 228-234.

19.Line 199-201: Figure 3C does not provide additional information to the text, remove Fig. 3A.

Response: Changed.

  1. Line 207: Correct spelling of “therefore”

Response: Changed.

  1. Line 211: The figures and the text within the figures is too small. Please make figures and text bigger.

Response: Changed.

22.Line 212: As Figure legends should be readable without the main text, please open the abbreviation DEM in the figure legend.

Response: Changed.

23.Line 224: Please make the circles and the text bigger as it is impossible to read the gene names from the figure

Response: Changed.

24.Line 225-228: Please also explain what the yellow colour stands for

Response: The yellow ellipses represent pigment genes. We have changed in the figure legend. Please see lines 270-271.

25.Line 242: Font for the miRNA gene names seems bigger than in the other text, please correct

Response: Changed.

  1. Line 260-262: In usually the EDNRB2 gene is the one with mutations associated plumage color, also in the references you have cited are about the EDBRB2 gene, not EDNRB (these are different paralogous genes). Also in online mendelian inheritance of animals (OMIA), I havent’t found any avian species with mutations reported in EDNBR, all mutations have been in EDNRB2 gene. Could you check that the gene meant here is really EDNRB

Response: We are greatly appreciate your valuable advice. Due to the fact that the Anser cygnoides reference genome (GCF_000971095.1) we used only contains annotated genes for EDNRB, upon thorough examination and sequence comparison, it was determined that they are indeed the same gene. To maintain consistency in nomenclature, we have opted to rename EDNRB as EDNRB2 in the revised manuscript.

27.Line 287: MLANA has been shown to be associated with white plumage in greylag geese (Yang et al. 2022 Using comparative genomics to detect mutations regulating plumage variations in graylag (A. anser) and swan geese (A. cygnoides). Gene 834:146612) and to almond color in domestic pidgeons (Bruders et al. (2020) A copy number variant is associated with a spectrum of pigmentation patterns in the rock pigeon (Columba livia). PLoS Genet 16(5): e1008274.

Response: Changed. Please see lines 335-336.

28.Line 289: SLC45A2 mutation known to affect feather coloration in chicken and quail (Gunnarsson 2007 Mutations in SLC45A2 Cause Plumage Color Variation in Chicken and Japanese Quail. Genetics 175: 867–877)

Response: Changed. Please see lines 339.

29.Line 307: Could be pointed out here that in future it would be interesting to genotype the Shitou geese for the 14-bp insertion in the EDNRB2 gene to see if the same mutation identified in several other Chinese domestic goose breeds is behind the white plumage also in Shitou geese

Response: Thanks to your valuable advice. We have added additional sentences in lines 366-371.

30.Supplementary materials. Please provide better table titles which explain what is in each column of the tables or change column titles to be more informative. For example ST-G4_count  is not informative, what is ST and G4. Or G4_TPM, what is G4 and what is TPM? avoid abbreviation without explaining them.

Response: We are grateful to these suggestions. We have changed the table and column titles of supplementary tables and added explanations of some abbreviations. Pleas see lines 432-439. The revised supplementary tables will be resubmitted.

Round 2

Reviewer 3 Report

Comments and Suggestions for Authors

Manuscript by Guo et al. has improved from the previous version and the comments were implemented in the manuscript. I have a few additional comments about spelling errors that I noticed, font size in some of the figures and a comment about the EDNRB/EDNRB2 issue. Detailed comments line-by-line below:

Line 40 Keywords: Correct spelling for Anser cygnoides (now incorrect). Remove microRNA (in the title already)

Line 87-88: The new additions were confusing as it now reads that there should be a male and a female grey-feathered goose and a male and a female white-feathered goose, but there is only two geese, please correct

Line 124: Change ”reversly” to ”reverse”

Line 136: Remove extra spaces between ”manufacturer’” and ”s instructions”

Line 149: Check if the is extra spaces in this sentence

Line 222 Fig. 2: Part A was still left in this version, remember to remove the A part of the figure. Also increase the font size within the volcano plot

Line 249-252: Explanation for part C and D is missing. Increase the font sizes in the figures, now too small to be read

Line 291: About the EDNRB2/EDNRB issue. I have found this a bit puzzling because previously I found EDNRB2 from the swan goose genome (Zhedong white), but just few weeks ago I could not found it anymore from the reference sequence. It is named differently in the current reference sequence of Sichuan White. EDNRB2 with the 14-bp insertion was now named as LOC106047519 endothelin receptor type B-like (Gene ID: 106047519) Accession number NW_025927715 REGION: 224517..235393.

We compared the sequences of EDNRB and LOC106047519 (EDNRB2) and these are different genes. See also Liu et al. 2019 Endothelins (EDN1, EDN2, EDN3) and their receptors (EDNRA, EDNRB, EDNRB2) in chickens: Functional analysis and tissue distribution. General and Comparative Endocrinology 283: 113231. Based on your comments, I understood that you have EDNRB, actually. So it has to be changed back to EDNRB. But it should be now noted in your manuscript that usually the plumage color polymorphism are reported in the EDNRB2 gene.

Comments on the Quality of English Language

Few writing mistakes were spotted and these were noted in the comments to authors

Author Response

Dear Reviewer 

We are grateful to your comments and valuable suggestions on our manuscript. These comments will assist us in further improving of this paper. We have made revisions according to these comments. All revisions are indicated in red in the manuscript. And we replaced new figures in the revised manuscript. We sincerely hope that these revisions would satisfy you. The point-to-point responses to the comments are shown as follows. 

Best regards,

Dr. Zhongping Wu

-----------------------------

The following is a point-to-point response to your comments.

Comments:

1.Manuscript by Guo et al. has improved from the previous version and the comments were implemented in the manuscript. I have a few additional comments about spelling errors that I noticed, font size in some of the figures and a comment about the EDNRB/EDNRB2 issue. Detailed comments line-by-line below:

Response 1: Thanks for this comment.

2.Line 40 Keywords: Correct spelling for Anser cygnoides (now incorrect). Remove microRNA (in the title already)

Response 2: Changed.

3.Line 87-88: The new additions were confusing as it now reads that there should be a male and a female grey-feathered goose and a male and a female white-feathered goose, but there is only two geese, please correct

Response 3: Sorry. We revised the figure according previous comments, but forgot to replace in the revised manuscript. We changed to new figures in the revised manuscript.

4.Line 124: Change ”reversly” to ”reverse”

Response 4: Changed.

5.Line 136: Remove extra spaces between ”manufacturer’” and ”s instructions”

Response 5: Changed.

6.Line 149: Check if the is extra spaces in this sentence

Response 6: Changed.

7.Line 222 Fig. 2: Part A was still left in this version, remember to remove the A part of the figure. Also increase the font size within the volcano plot

Response 7: See Response 3.

8.Line 249-252: Explanation for part C and D is missing. Increase the font sizes in the figures, now too small to be read

Response 8: See Response 3.

9.Line 291: About the EDNRB2/EDNRB issue. I have found this a bit puzzling because previously I found EDNRB2 from the swan goose genome (Zhedong white), but just few weeks ago I could not found it anymore from the reference sequence. It is named differently in the current reference sequence of Sichuan White. EDNRB2 with the 14-bp insertion was now named as LOC106047519 endothelin receptor type B-like (Gene ID: 106047519) Accession number NW_025927715 REGION: 224517..235393.

We compared the sequences of EDNRB and LOC106047519 (EDNRB2) and these are different genes. See also Liu et al. 2019 Endothelins (EDN1, EDN2, EDN3) and their receptors (EDNRA, EDNRB, EDNRB2) in chickens: Functional analysis and tissue distribution. General and Comparative Endocrinology 283: 113231. Based on your comments, I understood that you have EDNRB, actually. So it has to be changed back to EDNRB. But it should be now noted in your manuscript that usually the plumage color polymorphism are reported in the EDNRB2 gene.

Response 9: We are grateful to these comments once again. We checked the results of our RNA-seq analysis again and found that the ID of the screened differential mRNA was ncbi_106047519 (first column of the re-submitted table S2), which is consistent with the gene ID of LOC106047519 endothelin receptor type B-like (EDNRB2). However, the gene symbol was incorrectly labeled to EDNRB in our original manuscript. So, I still think it should be EDNRB2, and changed it’s full name to endothelin receptor type B-like in the revised manuscript. In addition, we extracted the sequence of LOC106047519 (placed in the attachment) from the previously downloaded reference genome of Zhedong White goose (GCF_000971095.1), which is currently unavailable on NCBI. And the sequence of LOC106047519 from Zhedong White goose differs from EDNRB in swan geese (Gene ID: 106043935, Accession number NW_025927833.1 REGION: 212675... 230009), which confirmed that they are indeed different genes. We also blasted the LOC106047519 transcript sequence (placed in the attachment) obtained from our mRNA sequencing with the NCBI database, and found it match to endothelin receptor type B-ike (LOC106047519) mRNA in Swan goose. This also confirms that our result is indeed EDNRB2.
